# The Relationship between the Unmet Needs of Chinese Family Caregivers and the Quality of Life of Childhood Cancer Patients Undergoing Inpatient Treatment: A Mediation Model through Caregiver Depression

**DOI:** 10.3390/ijerph191610193

**Published:** 2022-08-17

**Authors:** Jiamin Wang, Peter C. Coyte, Di Shao, Xuemei Zhen, Ni Zhao, Chen Sun, Xiaojie Sun

**Affiliations:** 1Centre for Health Management and Policy Research, School of Public Health, Cheeloo College of Medicine, Shandong University, Jinan 250012, China; 2NHC Key Lab of Health Economics and Policy Research, Shandong University, Jinan 250012, China; 3Institute of Health Policy, Management and Evaluation, Dalla Lana School of Public Health, University of Toronto, Toronto, ON M5T 2S8, Canada

**Keywords:** childhood cancer, family caregivers, depression, quality of life, inpatient treatment

## Abstract

A large proportion of the global burden of childhood cancer arises in China. These patients have a poor quality of life (QoL) and their family caregivers have high unmet needs. This paper examined the association between the unmet needs of family caregivers and the care recipient’s QoL. A total of 286 childhood cancer caregivers were included in this cross-sectional study. Unmet needs and depression among caregivers were assessed by the Comprehensive Needs Assessment Tool for Cancer Caregivers (CNAT-C) and the Patient Health Questionnaire (PHQ-9), respectively. The patient’s QoL was proxy-reported by the Pediatric Quality of Life Inventory Measurement Models (PedsQL 3.0 scale Cancer Module). Descriptive analyses, independent Student’s t-tests, one-way ANOVA, and mediation analyses were performed. The mean scores (standard deviations) for unmet needs, depression, and QoL were 65.47 (26.24), 9.87 (7.26), and 60.13 (22.12), respectively. A caregiver’s unmet needs (r = −0.272, *p* < 0.001) and depression (r = −0.279, *p* < 0.001) were negatively related to a care recipient’s QoL. Depression among caregivers played a mediating role in the relationship between a caregiver’s unmet needs and a care recipient’s QoL. As nursing interventions address depression among caregivers, it is important to standardize the programs that offer psychological support to caregivers.

## 1. Introduction

Childhood cancer is a global public health issue, and represents the second leading cause of death for children and adolescents and the sixth leading cause of the total cancer burden [1,2]. China shares a large proportion of the global burden of childhood cancer [3]. With growth in the incidence of childhood cancer, a decline in childhood cancer mortality, and relatively high survival rates in childhood cancer [4,5], China is facing an increasingly heavy disease burden. The COVID-19 pandemic has exacerbated this situation, as only one caregiver is permitted to accompany hospitalized patients. Confronted with this situation, family caregivers of childhood cancer patients who receive inpatient treatment have more unmet needs than they did at any previous juncture, potentially eroding the quality of life (QoL) of these patients. The purpose of this study was to assess the relationship between the unmet needs of Chinese family caregivers and the QoL of childhood cancer patients in receipt of inpatient treatment.

During hospitalization, childhood cancer patients endure both physical and physiological suffering [6]. Unmet needs are unresolved requirements that are necessary for prioritizing early interventions and health services to attain optimal well-being of cancer patients and their caregivers [7,8]. Aziza et al. studied the unmet needs of parents of childhood cancer patients in a hospital setting and demonstrated that 83% of parents had over ten unmet needs, such as the needs for information, psychological support, and work and social support [9]. Caregivers’ unmet needs may compromise the QoL of patients, as caregivers might have trouble in providing adequate care [10,11]. Wang and colleagues demonstrated the association between the unmet needs of caregivers and the QoL of patients [12]. While this causal link may hold for children, a recent Korean study addressing adult cancer patients was unable to find any relationship [8]. This finding may be due to the differential responses to family caregiving across patient groups and motivates the current study of childhood cancer patients.

Although an association between the unmet needs of family caregivers and the QoL of patients may be found, the factors accounting for this association are poorly understood. Sklenarova and colleagues demonstrated that many caregivers suffer anxiety and depression when patients are undergoing treatment [13]; Yang et al. found that unmet needs play a major role in generating negative psychological feelings among family caregivers [14]; and Heckel identified a positive correlation between the unmet needs of caregivers and their likelihood to suffer depression [15]. Depression among caregivers is also likely to be associated with the QoL of childhood cancer patients. Pierce et al. reported that the psychological problem of caregivers was a significant, independent predictor of poorer QoL in patients [16]. Furthermore, Borges and colleagues studied the burden of family caregivers and found that caregiver depression was also affected by the QoL of the patients [17]. Based on the above analysis, caregiver depression may act as a mediator of the relationship between the unmet needs of caregivers and the QoL of patients.

There is a paucity of research dealing with the relationship between the unmet needs of family caregivers and their impact on the QoL of care recipients in general, and in the context of childhood cancer treatment in particular. By addressing this relationship, we plan to add to the growing literature and will explore the mediating role of depression among caregivers in this relationship. In summary, there are three overarching research objectives: first, to measure the QoL of children with cancer and the unmet needs of their family caregivers; second, to assess the relationship between the unmet needs of family caregivers and the QoL of care recipients; and third, to explore the potential mediating role played by depression among caregivers in the relationship between the unmet needs of caregivers and the QoL of care recipients.

## 2. Materials and Methods

### 2.1. Study Population and Data Collection

This cross-sectional study was conducted in Jinan, Shandong Province, China, from May to August 2021 [18]. Shandong Province is an important coastal province in East China. Three regional hospitals for pediatric oncology treatment were included in the sampling frame. These comprised (1) a general hospital, (2) a specialized tumor hospital, and (3) a specialized pediatric hospital. Cluster sampling was conducted to ensure the recruitment of 149 patients with solid tumors from the specialized tumor hospital, 101 patients with hematological tumors from the general hospital, and, finally, 53 hematological tumor patients from the specialized pediatric hospital. While 359 potential patient–caregiver dyads were asked to participate, the sample of respondents was composed of 303 patient–caregiver dyads who offered consent and who were able to complete all data requirements, thereby yielding a response rate of 84.4%. Seventeen children who were less than 2 years of age were excluded to elicit proxy-reported measures of the QoL of patients using the Pediatric Quality of Life Inventory Measurement Models (PedsQL 3.0 scale Cancer Module). The resulting analysis sample comprised 286 patient–caregiver dyads. A study flowchart of participant selection is shown in Figure 1. Data were collected through face-to-face interviews. Eligible caregiver respondents were interviewed by trained postgraduates. Childhood cancer patients aged 2 to 14 years comprised the study population. Family caregivers of patients were informed about the study and were asked for their consent to participate. The children required a malignant tumor diagnosis and had to be undergoing treatment for their condition during the study period. The family caregivers who refused to consent and those who consented but were unable to participate due to a serious developmental, audiovisual, or cognitive impairment were excluded.

### 2.2. Conceptual Framework

The conceptual framework that guides our empirical work is outlined in Figure 2. Causation is hypothesized to flow from the unmet needs of family caregivers to the QoL of childhood cancer care recipients, with an expected positive correlation between these constructs. The unmet needs of caregivers, in turn, depend on an array of both caregiver and care recipient characteristics. These characteristics are also important in their potential direct effect on the QoL of patients. While the unmet needs of family caregivers may have a direct effect on the QoL of care recipients, our study is designed to also assess indirect effects through caregiver depression, thereby allowing for a possible mediating pathway of influence. The literature has explored this formulation, and in this study, we assess these possible relationships empirically.

### 2.3. Exposure and Outcome Assessment

#### 2.3.1. Exposure Variable

These unmet needs were assessed using the Comprehensive Needs Assessment Tool for Cancer Caregivers (CNAT-C) developed by Shin et al. [19]. CNAT-C is a valid and reliable needs-assessment tool involving 7 dimensions and 41 items, with each item measured on a 4-point Likert scale from 0 to 3. The higher the score, the greater the unmet needs. The Chinese version of this scale has high reliability and validity, with a Cronbach alpha of 0.81 [20].

#### 2.3.2. Mediator Variable

The Patient Health Questionnaire (PHQ-9) is a self-reporting questionnaire that consists of nine items for assessing symptoms of depression [21]. The score of each item ranges from 0 to 3, with a summed score ranging from 0 to 27 for the nine items. Its Chinese version has shown good reliability and validity, with a Cronbach alpha of 0.94 [22].

#### 2.3.3. Outcome Variable

PedsQL was developed by James W. Varni in 1987 [23]. The standard versions of the PedsQL 3.0 scale Cancer Module for Toddlers (2–4 years of age), Young Child (5–7 years of age), Child (8–12 years of age), and Adolescent (13–18 years of age) were adopted to assess the following eight dimensions (27 items): pain and hurt (2 items), nausea (5 items), treatment anxiety (3 items), procedural anxiety (3 items), worry (3 items), perceived physical appearance (3 items), cognitive problems (5 items), and communication (3 items). Five-level grades were used for scoring, ranging from 0 to 4. The score from the questionnaire was converted into a standard score by the item reverse-scoring method [23,24]. The total score and the score for each dimension ranged from 0 to 100. A higher score indicates a higher quality of life. The Chinese version of this scale has qualified reliability and validity, with a Cronbach alpha of 0.78 [24].

### 2.4. Statistical Analysis

Descriptive analyses were used to describe the sociodemographic characteristics of the childhood cancer patients and family caregivers, and the mean scores from different dimensions of unmet needs and QoL. An independent Student’s t-test and one-way ANOVA were used to compare the unmet needs of caregivers and the QoL of patients across different subgroups. We used multiple linear regression models (i.e., Model 1) to examine the association between the unmet needs of caregivers and the QoL of patients.

To test the potential indirect effects of depression among caregivers on the relationship between the unmet needs of caregivers and the QoL of patients, mediation analyses were performed. The relationships among the variables were examined by Pearson correlation analysis. The total effect of the unmet needs of caregivers on the patient’s QoL (c path), the association between the unmet needs of caregivers and the potential mediator variable depression among caregivers (a path), the independent effects of the mediator variable depression among caregivers (b path), and the exposure (c’ path; direct effect) of the outcome variable, the QoL of patients, were all tested through the use of multiple linear regression models (i.e., Models 2 to 4). Evidence for mediation was supported by the attenuation of the direct effect in comparison to the total effect, which was calculated as the difference between the total and direct effects (c–c’). Additionally, a 1000-sample bootstrap procedure was used to estimate the bias-corrected 95% confidence intervals (CIs) to test the significance of the indirect effect of the measured relationships. Evidence for the mediating effect was indicated when the 95% CIs of the indirect effect did not cross the zero line. All statistical tests were two-sided (*p* < 0.05) and were performed using Stata 15.0 (Stata Corp., College Station, TX, USA).

## 3. Results

### 3.1. Characteristics of Patients and Caregivers

The characteristics of the 286 childhood cancer patients and their caregivers are presented in Table 1. Among all childhood cancer patients, slightly more patients were diagnosed with leukemia (52.45%) than other cancers. The average ages for leukemia and Solid Tumor patients were 6.83 (3.39) and 6.25 (3.22) years, respectively, with similar proportions of female patients in each group. For the patients’ characteristics related to cancer treatment, Solid Tumor patients tended to arise more often among single-child households (*p* < 0.05), experienced more surgery (χ^2^ = 211.5011, *p* < 0.001) and suffered from recurrence more often (*p* < 0.001), were more likely to receive radiation therapy (*p* < 0.001), and tended to have more metastatic cancer (*p* < 0.001). For the caregivers of childhood cancer patients, the caregivers of leukemia patients were significantly older than those for Solid Tumor patients (*p* < 0.001), and 70.98% of all caregivers were women. The caregivers for Solid Tumor patients were also more likely to attain higher levels of education (*p* < 0.05) than those for leukemia patients.

### 3.2. Unmet Needs of Caregivers and the QoL for Care Recipients

Figure 3 demonstrates that the mean scores (and associated standard deviation, SD) for the QoL of patients and the unmet needs and depression of caregivers were 60.13 (22.12), 65.47 (26.24), and 9.87 (7.26), respectively. The dimensions resulting in a lower QoL for patients were procedural anxiety (52.16 ± 30.06), treatment anxiety (53.89 ± 27.95), and nausea (56.14 ± 26.37). Table 2 reports that the scores for QoL in the procedural anxiety dimension were significantly different for male and female caregivers (t = 1.87, *p* < 0.05), with those care recipients who had a female caregiver (mean = 50.04, SD = 30.01) reporting lower QoL scores in the procedural anxiety dimension. The QoL scores of solid tumor patients were lower in several dimensions, including pain and hurt (t = 1.81, *p* < 0.05), cognitive problems (t = 1.76, *p* < 0.05), procedural anxiety (t = 2.39, *p* < 0.01), perceived physical appearance (t = 4.11, *p* < 0.001), treatment anxiety (t = 4.34, *p* < 0.001), and nausea (t = 4.77, *p* < 0.001), than those for leukemia patients.

The dimensions with higher unmet needs for childhood cancer caregivers included healthcare staff (73.04 ± 30.31), information (67.20 ± 26.01), and hospital facilities and services (56.24 ± 29.36). These informational needs included treatment, dietary, home care, communication skills, and stress management. The healthcare staff needs included needs for timely visits, self-esteem, clear explanations about the disease, and good communication with healthcare providers. The informational needs included information about treatments, diet, home care, communication skills, and stress management.

Caregivers for leukemia patients tended to report lower levels of depression than their counterparts for solid tumor patients. Specifically, the last table reported in Figure 3 demonstrates that caregivers for solid tumor patients compared with those for leukemia patients were more common in the moderate to severe categories of depression. The opposite was the case for mild and minimal categories of depression.

### 3.3. Moderating Effect of Depression on the Association between Unmet Needs and QoL

Table 3 shows the correlation matrix for the three key study variables. All study variables were significantly correlated with each other (*p* < 0.001). The unmet needs of caregivers (r = −0.272) and depression reported by caregivers (r = −0.279) were negatively related to the QoL of patients. When each of those scores was higher, childhood cancer patients were more likely to report lower QoL. The unmet needs of caregivers were positively correlated with depression reported by caregivers (r = 0.291, *p* < 0.001).

The relationships among unmet needs and depression of caregivers and the QoL of care recipients are shown in Table 4. All models controlled for an array of covariates. In Model 1, the independent variable, unmet needs of caregivers, had a significant negative impact (β = −0.23, *p* < 0.001, R^2^ = 0.1846) on the dependent variable, QoL of patients. Model 2 demonstrated that the unmet needs of caregivers were positively and significantly (β = 0.11, *p* < 0.001, R^2^ = 0.2406) associated with the potential mediator variable, depression among caregivers. In Model 3, the mediator variable, caregiver’s depression, was significantly and negatively related (β = −0.11, *p* < 0.001, R^2^ = 0.2062) to the dependent variable, the unmet needs of caregivers. Finally, the independent variable, unmet needs of caregivers, and the mediator variable, caregiver’s depression, were jointly included in Model 4. The results showed that, while the unmet needs of caregivers were still significantly and negatively related (β = −0.14, *p* < 0.007) to the dependent variable, QoL of patients, this relationship was weaker when compared with Model 1, as the mediator variable, caregiver’s depression, played a significant mediating effect (Appendix A).

Table 5 shows the total effect of the unmet needs of caregivers on the QoL of patients, which comprises both the direct and indirect effects. The indirect effect that was channeled through depression among caregivers was statistically significant (β = −0.087, 95% CI −0.136 to −0.041, *p* < 0.001), with the mediating effect accounting for 38.16% of the total effect. Therefore, some of the association between the unmet needs of caregivers and the QoL of patients was achieved through the depression mediator variable.

## 4. Discussion

### 4.1. Main Findings

This study showed the mediating role of depression among caregivers in channeling the impact of the unmet needs of caregivers to the QoL of patients, and therefore provides insights into mechanisms that can be used to foster improvements in the QoL of childhood cancer patients while they are undergoing inpatient therapy and receiving support from family caregivers.

### 4.2. Level of QoL among Chinese Childhood Cancer Patients

Our findings may be placed in the context of the literature. The mean (and SD) score for the QoL for childhood cancer patients was in line with those in studies conducted in Turkey (mean = 64.51, SD = 18.27) [25] and Brazil (mean = 62.90, SD = 16.6) [26,27]. Our QoL scores were slightly lower compared with those of other studies, as our study included more patients with solid tumors (47.55%) than those reported in the literature [28]. Ji and colleagues reported a smaller proportion (38.70%) of solid tumor patients in their study, and they reported higher QoL scores (mean = 74.03, SD = 22.54) than our findings [24]. Unlike leukemias, solid tumors may present with vaguer symptoms, cause diagnosis delays, display the effects of loco-regional diseases, suffer from more intensive therapy, and more negative prognosis outcomes [29]. Ho and colleagues compared the QoL of childhood solid tumor and leukemia patients, and demonstrated that the QoL scores were lower for children with solid tumors than those with leukemia [30].

In our study, female caregivers were significantly affected by the patient’s procedural anxiety scale score, with procedural anxiety for women being greater than that reported by male caregivers. Procedural anxiety includes fear and pain from needles and blood tests. Similar results were also reported by Zheng and colleagues, and these results demonstrate the importance of the caregivers’ gender in the anxiety of patients [31]. This might be due to the traditional role of Chinese mothers. Many mothers may feel guilty for their young children’s illness and blame themselves for the occurrence of cancer, thereby raising their levels of anxiety [26]. These negative emotions may be passed on to their children through daily interaction.

### 4.3. Relationship between Unmet Needs of Caregivers and the QoL of Patients

The current study found that the unmet needs of caregivers were negatively associated with the QoL of patients. The study by Nicklin et al. illustrated the negative association between the unmet needs of caregivers and the QoL of both patients and caregivers [32]. The autonomy of children who are highly dependent on their caregivers is poor, especially in the twin areas of communication with medical staff and treatment decision-making. Consequently, when the needs of family caregivers were not met, this may signal insufficient support for children that, in turn, results in a decline in the QoL of childhood cancer patients. Opportunities to advance the QoL of childhood cancer patients would be gained by responding to and more completely satisfying the unmet needs of family caregivers.

### 4.4. Moderating Role of Caregiver’s Depression in the Relationship between the Unmet Needs of Caregivers and the QoL of Patients

To our knowledge, this study is the first to examine the mediating effect of depression among caregivers on the association between the unmet needs of caregivers and the QoL of patients. Previous studies have shown a positive association between the unmet needs of caregivers and depression among family caregivers [15]. Furthermore, it has been shown that the emotional status of caregivers is an important risk factor for the QoL of patients [33]. The psychological well-being of family caregivers is integral to the provision of quality care and important to improving recovery [34]. The prevalence of depression has been shown to be higher among family caregivers when cancer patients are in receipt of treatment [13]. Yang et al. focused on the unmet needs and emotional states of caregivers in the inpatient setting and found that the unmet needs of caregivers had a major effect on the depression of caregivers when the patient was undergoing treatment [14].

Furthermore, the high unmet informational needs demonstrated that most family caregivers were not prepared to provide care, and had little knowledge about the disease and treatments undertaken. Combined with the high unmet healthcare staff needs and unmet needs for hospital facilities and services, we infer that China’s pediatric resources are relatively scarce, and there is a mismatch between the current state of medical care and the needs of family caregivers. Zhang et al. also highlighted the unmet needs for pediatric care and stressed that it represents an important challenge facing China’s pediatric health care system [27].

### 4.5. Study Limitations

There are several study limitations. First, this study used a relatively small sample and was conducted in mainland China, so opportunities to generalize the findings may be limited. This limitation may be attenuated somewhat, as several similar studies have demonstrated cross-cultural success [19,35,36]. Second, we only examined parent proxy-reported QoL without measuring child self-reported QoL directly, as the children in this study were either too young or too weak to report themselves. Previous studies have shown good concordance between parent proxy reporting and self-reporting for the PedsQL 3.0 module, which suggests that the included measures may be useful approximations for the QoL of the included children [22]. Although child self-reporting is important, family perspectives are also important, as described previously. Third, as our study was cross-sectional, only associations were revealed, rather than definitive causal relationships. However, the correlations reported between the study variables were both consistent with the existing literature and in line with the theoretical model that guided our empirical study. Fourth, while depression may vary over time, our study only assessed depression at a single point in time. The adoption of a longitudinal study is needed to explore the long-term impact of the unmet needs of caregivers on the potential time-varying course of the QoL of patients via the time-varying course of depression among caregivers.

## 5. Conclusions

This study outlined mechanisms of influence from the unmet needs of family caregivers of children undergoing inpatient treatment for cancer and its deleterious effect on the QoL of patients. The unmet needs of caregivers were shown to be associated with an erosion in the QoL of childhood cancer patients by mediating the depression of caregivers. The recognition of these relationships provides insights into mechanisms that might be used to enhance the QoL of patients. Indeed, interventions and supports, such as tailored psychological screening programs and counseling services, designed to anticipate the unmet needs of caregivers may help to reduce the incidence of depression among caregivers and offer both direct and indirect channels through which the QoL of childhood cancer patients may be advanced.

## Figures and Tables

**Figure 1 ijerph-19-10193-f001:**
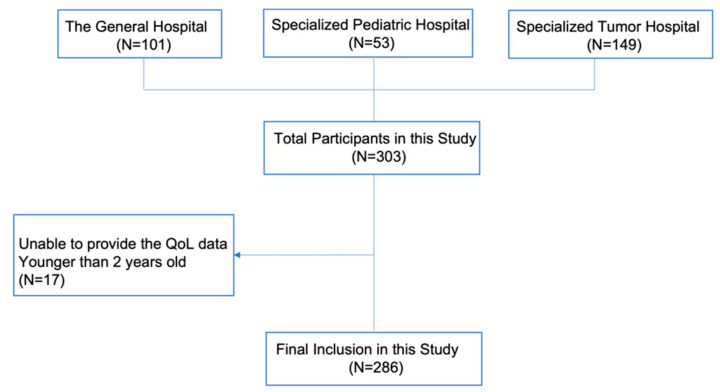
Study flowchart of participant selection.

**Figure 2 ijerph-19-10193-f002:**
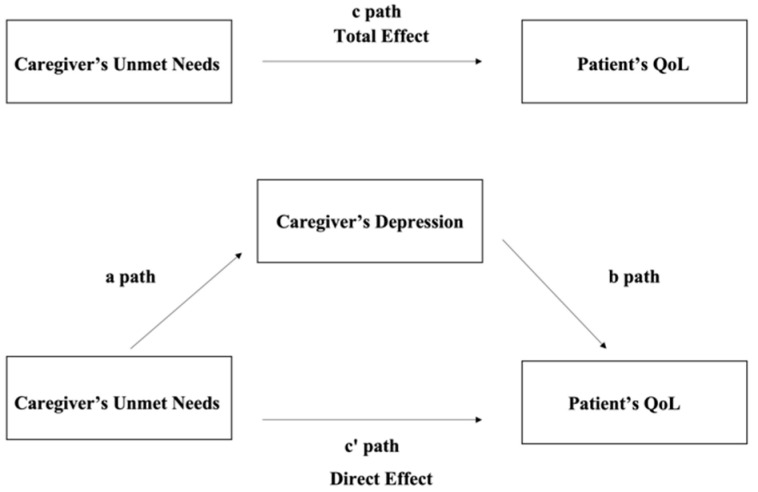
Hypothesized model.

**Figure 3 ijerph-19-10193-f003:**
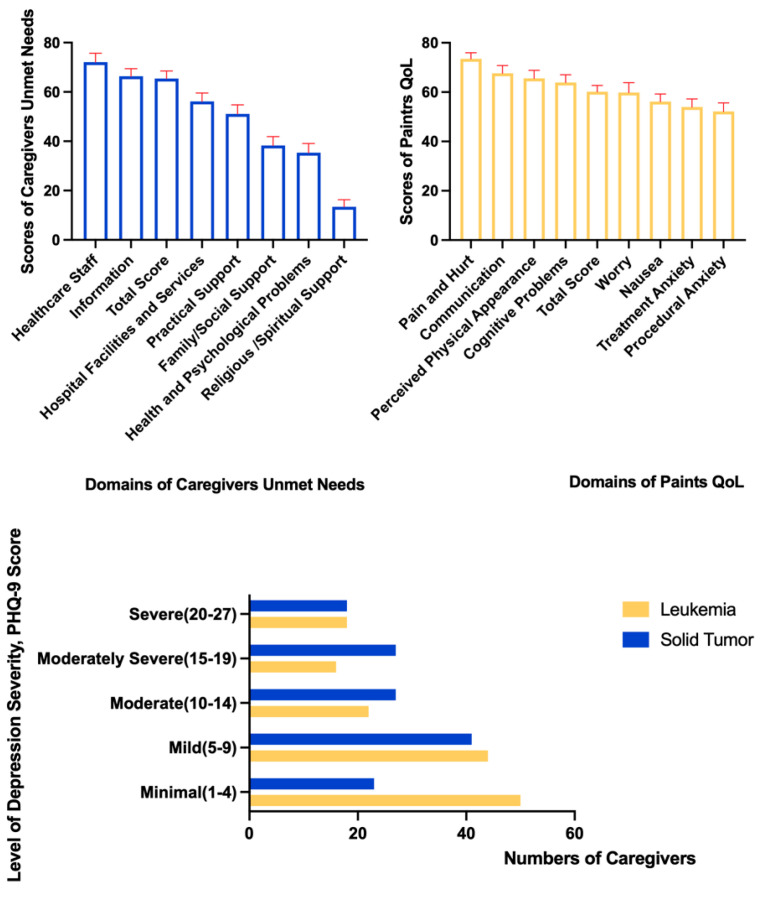
Descriptive statistics for the unmet needs and depression of caregivers and the QoL of care recipients (N = 286).

**Table 1 ijerph-19-10193-t001:** Characteristics of childhood cancer patients and their caregivers based on cancer type (N = 286).

Characteristics	Cancer Type		
	Leukemia (N = 150, 52.45%)	Solid Tumor (N = 136, 47.55%)	*p*
* **Patients** *			
Age, mean (SD *)	6.83 (3.39)	6.25 (3.22)	0.1424
Gender			0.487
Male	91 (60.67)	77 (56.62)	
Female	59 (39.33)	59 (43.38)	
Single Child			
Yes	29 (19.33)	45 (33.09)	<0.05 *
No	121 (80.67)	91 (66.91)	
Surgery			<0.001 ***
Yes	8 (5.33)	124 (91.18)	
No	142 (94.67)	12 (8.82)	
Radiation therapy			<0.001 ***
Yes	2 (1.33)	75 (55.15)	
No	148 (98.67)	61 (44.85)	
Number of chemotherapy			0.718
≤10	74 (49.33)	70 (51.47)	
>10	76 (50.67)	66 (48.53)	
Recurrence			<0.001 ***
Yes	4 (2.67)	45 (33.09)	
No	146 (97.33)	91 (66.91)	
Metastasis			<0.001 ***
Yes	3 (2.00)	79 (58.09)	
No	147 (98.00)	57 (41.91)	
* **Caregivers** *			
Age, mean (SD *)	37.27 (6.12)	34.54 (5.29)	<0.001 ***
Gender			0.702
Male	45 (30.00)	38 (27.94)	
Female	105 (70.00)	98 (72.06)	
Education Status			<0.05 *
Compulsory Education	83 (55.33)	58 (42.65)	
High School and Above	67 (44.67)	78 (57.35)	

* *p* < 0.05, and *** *p* < 0.001.

**Table 2 ijerph-19-10193-t002:** Differences in QoL domains regarding the caregiver and cancer characteristics (N = 286).

Caregiver Characteristics	Pain and Hurt	Nausea	Procedural Anxiety	Treatment Anxiety	Worry	Cognitive Problems	Perceived Physical Appearance	Communication
Gender								
Male	74.25 ± 22.71	58.19 ± 25.96	57.33 ± 29.70	52.01 ± 29.09	53.92 ± 34.27	61.75 ± 28.66	59.54 ± 28.67	65.70 ± 25.97
Female	73.15 ± 21.32	55.30 ± 26.55	50.04 ± 30.01	54.66 ± 27.50	62.23 ± 34.41	64.74 ± 26.69	68.06 ± 27.02	68.42 ± 27.30
t	0.3865	0.8428	1.8694	−0.7264	−1.8574	−0.8415	−2.3785	−0.7734
*p*	0.3497	0.2000	0.0313 *	0.7659	0.9679	0.7996	0.9910	0.7800
Cancer Type								
Leukemia	75.67 ± 20.72	62.97 ± 25.18	56.17 ± 31.89	60.51 ± 28.75	69.95 ± 34.15	66.56 ± 27.00	71.83 ± 27.20	68.54 ± 27.42
Solid Tumor	71.05 ± 22.55	48.60 ± 25.68	47.73 ± 27.33	46.58 ± 25.18	58.58 ± 35.00	60.91 ± 27.34	58.70 ± 26.76	66.62 ± 26.39
t	1.8050	4.7726	2.3894	4.3393	0.5782	1.7564	4.1094	0.6042
*p*	0.0361 *	<0.001 ***	0.0088 **	<0.001 ***	0.2818	0.0400 *	<0.001 ***	0.2731

* *p*-value < 0.05; ** *p*-value < 0.01; *** *p*-value < 0.001.

**Table 3 ijerph-19-10193-t003:** Pearson correlations among the unmet needs and depression of caregivers and the QoL of Care recipients (N = 286).

Variables	Patient’s QoL	Caregiver’s Unmet Needs	Caregiver’s Depression
Patient’s QoL	1		
Caregiver’s Unmet Needs	−0.272 ***	1	
Caregiver’s Depression	−0.279 ***	0.291 ***	1

*** *p*-value < 0.001.

**Table 4 ijerph-19-10193-t004:** Multiple linear regression models among the unmet needs and depression of caregivers and the QoL of care recipients (N = 286).

Characteristics	Model 1 † (Patient’s QoL)				Model 2 † (Caregiver’s Depression)				Model 3 † (Patient’s QoL)				Model 4 † (Patient’s QoL)			
	***β*** (95% CI)	*p*	R^2^	F	***β*** (95% CI)	*p*	R^2^	F	***β*** (95% CI)	*p*	R^2^	F	***β*** (95% CI)	*p*	R^2^	F
			0.1846	3.17 ***			0.2406	4.44 ***			0.2062	3.64 ***			0.2405	4.19 ***
Depression									−0.11 (−0.15–−0.07)	<0.001			−0.84 (−1.21–−0.47)	<0.001		
Unmet Needs	−0.23 (−0.32–−0.13)	<0.001			0.11 (0.08–0.14)	<0.001							−0.14 (−0.24–−0.04)	0.007		

†: Adjusted for other cofounders: patients’ socioeconomic characteristics, patients’ cancer characteristics, and caregiver characteristics. *** *p*-value < 0.001.

**Table 5 ijerph-19-10193-t005:** Mediation analysis results.

Model Pathways	* **β** *	S.D.	95% CI	*p*	Mediating Effect
Total Effect I	−0.228	0.048	−0.324–−0.133	<0.001	100%
Direct Effect (c’)	−0.141	0.052	−0.237–−0.031	0.007	61.84%
Indirect Effect (c–c’)	−0.087	0.025	−0.136–−0.041	<0.001	38.16%
Path a	0.106	0.015	−0.076–−0.136	<0.001	\
Path b	−0.838	0.188	−1.207–−0.468	<0.001	\

Notes: QoL = Quality of Life; CI = Confidence Interval.

## Data Availability

The datasets used and analyzed in this study are available from the corresponding author upon reasonable request.

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
