# Peer review of "The Relationship between the Unmet Needs of Chinese Family Caregivers and the Quality of Life of Childhood Cancer Patients Undergoing Inpatient Treatment: A Mediation Model through Caregiver Depression"

_ijerph, 2022, doi:10.3390/ijerph191610193_

Round 1

Reviewer 1 Report

Dr. Jiamin Wang and co-authors conducted a cross-sectional study on the mediation role of depression in childhood cancer’s caregivers between their unmet needs and the quality of life of patients.

The topic is relevant for public health and interesting. Anyway, the study has some limitations.

Major Criticisms:

  • Pg. 2, (Introduction): Maybe a definition of “unmet needs” in oncologic field would be useful.

  • Results: Caregivers’ depression scores (Patient Health Questionnaire, PHQ-9) are missing.

  • Results, pg. 5 (paragraph 3.2): PedsQL 3.0 and CNAT-C are different instruments that evaluate different constructs (quality of life of patients and unmet needs of caregivers). The presentation of the results of these test seems to make a comparison, and results misleading. Please, presented it separately to avoid possible wrong interpretations.

  • Discussion, pg. 9 (paragraph 4.2): Please, provide an explanation as to why for Authors solid tumor patients have worse QoL compared with Leukemia patients.

  • The manuscript should be revised by a native English speaking.

Minor criticism:

  • Pg. 1, Abstract: Please, add the instruments used to assess caregivers’ unmet needs (Comprehensive Needs Assessment Tool 117 for Cancer Caregivers CNAT-C) and depression (Patient Health Questionnaire PHQ-9).

  • Pg. 2 lines 45-46: “…and demonstrated that 83% of parents had over ten unmet 45 needs”. Please, indicate what these unmet needs refer to.

  • Pg. 2 lines 79-80: “Shandong province, one of the important coastal provinces in 79 East China “A verb seems to be missing in this sentence.

  • Pg. 3 line 103: Maybe the Authors are referring to Figure 2 (not Figure 1).

  • Pg. 3 figure 2: The model could be easier to read if in the boxes Authors add “Caregivers’ unmet needs”, “Caregivers’ depression” and “Patients’ QoL”.

  • Pg. 5 line 177: Maybe the Authors are referring to Figure 3 (not Figure 2).

  • Pg. 5 line 177: “Demonstrated” is too strong, maybe “showed” could be better.

  • Pg. 6 line 194: The description of the figure refers to Figure 3 (not Figure 2).

  • Pg. 9 line 228: “Demonstrated” is too strong, maybe “showed” could be better.

  • Pg. 9 lines 236-238: “Our QoL scores 236 were slightly lower compared with other studies potentially because we include more 237 patients of solid tumor…”. Please, clarify this sentence.

Reviewer 2 Report

General comments

I congratulate the Authors on this genuine and practical study. I read the manuscript with interest and pleasure. I recognize the manuscript as well-prepared and the study as decently conducted.  I like the study design. The recruitment is consecutive, the methodology has a decent background and provides an easy-to-follow overview of enrollment and evaluation. The hypothesis is clear and provides a sufficient rationale for conducting the study. Statistical evaluation seems genuine. The mediation analysis (which strictly corresponds with the hypothesis) was implemented to determine the indirect effects. That is not always present in comparable studies. A bootstrapping simulation was utilized for the internal validation of outcomes.

Minor issues

- minor language review would be appreciated, the manuscript contains punctuation (e.g. overuse of semicolons) and spelling errors (e.g. ‘phycological’ line 57); in particular, the section Results requires linguistic corrections - considering both style and grammar

- I would consider changing X1, X2 and X3 in the correlation matrix to the full names of the variables, I feel it would be more intuitive to read;

- I would eventually consider supplementing the table which presents multivariable regression models with adjustment variables (coefficients and p’s) – but it is up to the Authors; my impression is that it provides deeper insight into confounding to Reader (supplementary file?), but again I understand not putting it here.
